# RDAG U-Net: An Advanced AI Model for Efficient and Accurate CT Scan Analysis of SARS-CoV-2 Pneumonia Lesions

**DOI:** 10.3390/diagnostics14182099

**Published:** 2024-09-23

**Authors:** Chih-Hui Lee, Cheng-Tang Pan, Ming-Chan Lee, Chih-Hsuan Wang, Chun-Yung Chang, Yow-Ling Shiue

**Affiliations:** 1Institute of Biomedical Sciences, National Sun Yat-sen University, Kaohsiung 804, Taiwan; julia@mstc.com.tw; 2Department of Mechanical and Electro-Mechanical Engineering, National Sun Yat-sen University, Kaohsiung 804, Taiwan; pan@mem.nsysu.edu.tw; 3Institute of Advanced Semiconductor Packaging and Testing, College of Semiconductor and Advanced Technology Research, National Sun Yat-sen University, Kaohsiung 804, Taiwan; 4Institute of Precision Medicine, National Sun Yat-sen University, Kaohsiung 804, Taiwan; 5Taiwan Instrument Research Institute, National Applied Research Laboratories, Hsinchu City 300, Taiwan; 6Department of Electrical Engineering, National Kaohsiung University of Science and Technology, Kaohsiung 807, Taiwan; mclee@nkust.edu.tw; 7Nephrology and Metabolism Division, Department of Internal Medicine, Kaohsiung Armed Forces General Hospital, Kaohsiung 802, Taiwan; wangchihhsuan@gmail.com; 8Institute of Medical Science and Technology, National Sun Yat-sen University, Kaohsiung 804, Taiwan

**Keywords:** U-Net, image recognition, computed tomography, pneumonia, 3D segmentation

## Abstract

**Background/Objective:** This study aims to utilize advanced artificial intelligence (AI) image recog-nition technologies to establish a robust system for identifying features in lung computed tomog-raphy (CT) scans, thereby detecting respiratory infections such as SARS-CoV-2 pneumonia. Spe-cifically, the research focuses on developing a new model called Residual-Dense-Attention Gates U-Net (RDAG U-Net) to improve accuracy and efficiency in identification. **Methods**: This study employed Attention U-Net, Attention Res U-Net, and the newly developed RDAG U-Net model. RDAG U-Net extends the U-Net architecture by incorporating ResBlock and DenseBlock modules in the encoder to retain training parameters and reduce computation time. The training dataset in-cludes 3,520 CT scans from an open database, augmented to 10,560 samples through data en-hancement techniques. The research also focused on optimizing convolutional architectures, image preprocessing, interpolation methods, data management, and extensive fine-tuning of training parameters and neural network modules. **Result:** The RDAG U-Net model achieved an outstanding accuracy of 93.29% in identifying pulmonary lesions, with a 45% reduction in computation time compared to other models. The study demonstrated that RDAG U-Net performed stably during training and exhibited good generalization capability by evaluating loss values, model-predicted lesion annotations, and validation-epoch curves. Furthermore, using ITK-Snap to convert 2D pre-dictions into 3D lung and lesion segmentation models, the results delineated lesion contours, en-hancing interpretability. **Conclusion:** The RDAG U-Net model showed significant improvements in accuracy and efficiency in the analysis of CT images for SARS-CoV-2 pneumonia, achieving a 93.29% recognition accuracy and reducing computation time by 45% compared to other models. These results indicate the potential of the RDAG U-Net model in clinical applications, as it can accelerate the detection of pulmonary lesions and effectively enhance diagnostic accuracy. Additionally, the 2D and 3D visualization results allow physicians to understand lesions' morphology and distribution better, strengthening decision support capabilities and providing valuable medical diagnosis and treatment planning tools.

## 1. Introduction

The Severe Acute Respiratory Syndrome Coronavirus (SARS-CoV) was first isolated and identified in 2003, marking the onset of the initial pandemic caused by a coronavirus. Subsequently, the Middle East Respiratory Syndrome (MERS) coronavirus emerged, becoming the deadliest coronavirus-related disease to date. In 2020, the pneumonia caused by SARS-CoV-2, commonly referred to as COVID-19, escalated into a global respiratory pandemic, overwhelming healthcare systems worldwide. According to data from the World Health Organization [1], as of September 2022, the cumulative global confirmed cases reached 609 million, resulting in 6,500,000 deaths. The ongoing threat of respiratory infections continues to pose a significant global health crisis, placing immense stress on healthcare infrastructures and necessitating continuous research into effective treatment strategies. The persistent challenge of respiratory infections, including COVID-19, highlights the critical need for advanced diagnostic radiological techniques to facilitate early intervention and improve patient outcomes. The primary diagnostic method for COVID-19 has been reverse transcription polymerase chain reaction (RT-PCR) testing [2,3]. Although RT-PCR is the gold standard for detecting SARS-CoV-2, it is labor-intensive, time-consuming, and exposes healthcare workers to potential infection risks. These limitations underscore the importance of developing alternative diagnostic methods that are more efficient, faster, and safer for healthcare providers.

Patients infected with COVID-19 often exhibit Ground Glass Opacity (GGO) in their lungs, visible in CT scans. Artificial intelligence (AI) can be trained to recognize these GGOs, providing rapid and accurate analysis of chest CT images while awaiting RT-PCR results. This AI technology can perform preliminary screening, ruling out uninfected cases and identifying potential COVID-19 cases, thus alleviating the burden on healthcare personnel and reducing the workforce required for testing.

CT scans are used extensively for imaging various parts of the human body [4]. For lung imaging, CT scans offer higher resolution and sensitivity compared to magnetic resonance imaging (MRI) [5], and can be enhanced with contrast agents to visualize lesions not visible to the naked eye. Commonly used iodinated contrast media absorb X-ray energy, creating contrast by limiting X-ray penetration, which helps distinguish lesions from normal tissues. Although most of the contrast media are metabolized and excreted through urine, a kidney function test is necessary before injection due to potential side effects, including mild reactions such as warmth, nausea, and vomiting and severe allergic reactions in some individuals.

Artificial intelligence [6] aims to train computers to achieve cognitive and learning capabilities similar to humans. Deep learning [7], a sophisticated branch of AI, employs artificial neural networks to mimic the neural architecture of the human brain. These networks consist of an input layer, multiple hidden layers, and an output layer. The complexity of deep learning networks arises from the numerous hidden layers involved. The ultimate goal of AI and deep learning is to enable computers to approximate human cognitive abilities through complex neural networks.

The sudden onset of the global respiratory pandemic underscored the crucial role of chest CT scans in diagnosing emergency department patients. AI can assist radiologists in making diagnoses and determining treatments. Neural networks can be categorized into Convolutional Neural Networks (CNNs) [8] for image recognition and classification, and Fully Convolutional Networks (FCNs) [9] for image segmentation tasks. FCNs, with their capability for pixel-wise image segmentation, are more suitable for medical image visualization than CNNs.

Common neural network models for image segmentation include FCN, SegNet [10], U-Net++ [11], and U-Net [12]. The U-Net model, proposed by Olaf et al. [12], achieves excellent results with limited training data, making it ideal for medical image segmentation. Oktay et al. [13] introduced the Attention U-Net model for pancreas segmentation and Zhang et al. [14] enhanced U-Net with Atrous Spatial Pyramid Pooling (ASPP) and Attention modules for retinal vessel segmentation, achieving high accuracy. Nillmani et al. [15] combined U-Net with other classification models for COVID-19 applications, demonstrating superior performance. Lv et al. [16] developed TB-Net for bone cancer segmentation using a U-Net-based transformer, achieving high accuracy metrics. Wu et al. [17] proposed the Dual Kmax UX-Net, a semi-supervised deep learning model to address the issue of insufficient data labeling. Table 1 will present the features of these studies.

The U-Net architecture, with its convolution and deconvolution layers, forms a U-shaped design that excels in medical image segmentation tasks, even with small datasets. Its efficient structure leads to shorter training times compared to CNNs, making it a preferred choice for medical image recognition. Our research team has previously applied deep learning for AI-based diagnosis and prediction of cancerous lesions in organs such as the bladder [18] and liver [19].

The contributions of this research are:Improved training time and accuracy in lesion identification without the need for contrast agents;Identification of the most suitable model for pneumonia diagnosis among Attention Res U-Net, Attention U-Net, and Residual-Dense-Attention Gates U-Net (RDAG U-Net) through comparative analysis;Compared to the Attention Res U-Net and Attention U-Net models, RDAG U-Net converges to high accuracy more quickly, significantly shortening the training time (up to 45%).

This study highlights the potential of AI in transforming medical diagnostics, particularly in the context of the global respiratory pandemic.

## 2. Methods

We structured the operation of the neural network into three distinct modules to facilitate precise parameter control and mitigate the risk of parameter interaction effects, which could lead to biased observations. These modules are as follows:(1)Training parameters;(2)Module adjustments;(3)Model architecture.

Organizing the workflow into these three modules enables a systematic approach to fine-tuning and optimizing the neural network. This structured methodology facilitates the achievement of accurate and reliable results by allowing targeted adjustments and comprehensive evaluation of each module’s impact on overall performance.

### 2.1. Training Parameters

The training parameters encompass critical factors such as image resolution, batch size, number of epochs, interpolation method, and kernel size, all of which directly impact the training process. Careful adjustment of these parameters is necessary to balance hardware limitations, training duration, and model performance evaluation. Batch size, or training batch, refers to the number of training samples processed in a single iteration. A larger batch size can shorten training time and potentially enhance model robustness, subject to hardware constraints. However, it often has an inverse relationship with input image resolution. During training, images are frequently downscaled to save memory, but maintaining a higher resolution closer to the original image can capture more detailed and richer features. Therefore, increasing resolution to improve model recognition performance is one of our research directions. The trade-off between resolution and batch size is crucial in training, with batch size generally prioritized. Enhancing model performance by increasing resolution while maintaining batch size is also considered. This experimental parameter design is based on hardware testing, using the maximum batch size possible at different resolutions to ensure optimal model performance [20]. In deep learning model training, the trade-off between resolution and batch size is crucial. Batch size affects the stability and convergence speed of the model’s training. Larger batch sizes generally provide more stable gradient estimates, reduce noise during training, and thus help the model converge faster and more stably. However, larger batch sizes may lead to interruptions or significant reductions in efficiency due to memory constraints on limited hardware resources. Therefore, investigating how to improve resolution while maintaining batch size to enhance model performance is also an important direction. This experimental parameter design aims to find an optimal balance between model performance and computational resources, which can help achieve performance improvements while maintaining training efficiency by leveraging the benefits of higher resolution. Interpolation involves algorithms used for image compression, enlargement, and other processing techniques. To improve model recognition performance without increasing training image resolution, interpolation methods can reduce compression artifacts, preserving image features while maintaining the advantages of large batch sizes. Each interpolation method is domain-specific, so selecting the method that best suits the neural network’s recognition features is crucial [21]. The kernel size in convolutional layers significantly influences the learning of pixel relationships in neural networks. Larger kernel sizes capture more comprehensive pixel relationship information, resulting in more robust global features. However, this also increases the data processed by the neural network, leading to greater model complexity and larger storage requirements. To address this, we refer to the principle of using smaller stacked kernels as discussed in CS231n. For instance, stacking three 3 × 3 kernel convolutions achieves an effective kernel size of 7 × 7 while using only 55.1% of the parameters compared to a single 7 × 7 kernel. This approach enhances global feature strength and improves model accuracy without significantly increasing complexity. By carefully adjusting these training parameters, we aim to optimize the training process and achieve superior performance for the neural network model.

### 2.2. Data Set Creation and Image Processing

Data: The image dataset used in this study was obtained from an open-source website [22]. Blurry and hard-to-identify images were excluded, resulting in the selection of 20 patients confirmed to have COVID-19, comprising a total of 3520 CT images. To accommodate the model’s constraints, the original 512 × 512 resolution images were resized to 224 × 224 to reduce the number of pixels the model needs to process. This alleviates the computational burden, helping the model train or infer faster and enabling it to handle larger datasets or train under hardware constraints.

Data Augmentation: Acquiring a sufficient training set of medical images is often a significant challenge in medical image segmentation, limiting the generalizability of the training data. While U-Net neural networks perform well in overcoming such challenges, the diversity of training data is typically positively correlated with model performance. To address this issue, we incorporated a data augmentation module into the neural network by combining COVID-19 CT images obtained from the open-source website with our data augmentation techniques. By applying image processing techniques such as rotation and horizontal flipping, which alter the relative relationships between pixels, we added diversity to the training data [23].

HU Value: A Hounsfield Unit (HU) is a linear mapping of the measured linear attenuation coefficients of a medium, with distilled water and air at standard temperature and pressure set to 0 HU and −1000 HU, respectively. HU values are commonly used in X-ray CT scans and are also referred to as CT Numbers. In this study, due to the difficulty in identifying lesion areas in medical images without contrast agents, we adjusted the HU values of the original images through an image processing module to more prominently display the lung and lesion areas, thereby enhancing the accuracy of AI training [24].

Training/Validation Data Set Creation: Before creating the dataset, in addition to the previously mentioned HU adjustments and data augmentation, each CT image was paired with its corresponding lesion annotation to ensure consistency. The processed data were then randomly shuffled to mitigate overfitting. Subsequently, the data were split into training and validation sets in an 8:2 ratio, as illustrated in Figure 1.

By meticulously processing the dataset and employing advanced image augmentation techniques, we aimed to enhance the training process and improve the overall performance of the neural network model.

### 2.3. Evaluation Standards for Experimental Results

The experiment employs a univariate analysis approach, sequentially modifying different parts of the neural network, running experiments, recording observations, and making further experimental designs and hypotheses based on the results. Consequently, the interpretation and evaluation of experimental results are crucial. To more objectively assess the impact of model modifications, we have chosen the following evaluation criteria:Model Training Loss–Epochs/Accuracy–Epochs Curves:

Loss–Epochs Curve: The loss curve is plotted over the training epochs to help understand the model’s convergence during training. A significant drop in the loss curve indicates that the model is effectively learning and reducing errors, while a plateau or rise may suggest overfitting or other issues. This curve demonstrates whether the model is capable of minimizing errors throughout the training process.

Accuracy–Epochs Curve: The accuracy curve is plotted over the training epochs to provide insight into the model’s increasing accuracy. A gradually rising accuracy curve indicates an improvement in the model’s correct predictions. However, a plateau or decline may indicate overfitting or model underperformance.

Analyzing these curves allows us to make informed improvements to the model’s architecture and training process, thereby gaining a deeper understanding and enhancing model performance. The Loss–Epochs curve helps identify how well the model is learning, while the Accuracy–Epochs curve provides a measure of the model’s predictive performance over time. By examining these curves, we can adjust our approach to achieve optimal model performance and reliability.
2.Quantitative Metrics for Model Evaluation:

Evaluation metrics are crucial for assessing model performance. In this experiment, we will use the loss function [25] and several metrics derived from the confusion matrix [26] to evaluate the experimental results. The loss function calculation module is essential for assessing the discrepancy between the model’s predicted lesions and the actual lesion annotations. The loss value quantifies the degree of deviation between the model’s predictions and the ground truth. Depending on specific needs, various loss functions can be employed, such as cross-entropy loss [27], mean squared error (MSE) [28], or Dice loss [29].

By integrating the loss function calculation module, we can quantitatively assess the model’s performance, identify areas needing improvement, and fine-tune the model to achieve higher accuracy and lower loss during training. This module provides a robust framework to measure the efficacy of the model’s learning, enabling precise adjustments to enhance its predictive accuracy and overall performance.

The confusion matrix’s parameters, including True Positive (TP), False Positive (FP), True Negative (TN), and False Negative (FN), play a crucial role in this experiment. These four parameters form the evaluation metrics used in this experiment, which include accuracy, Intersection over Union (IOU), Dice Score Coefficient (DSC), and Average Hausdorff Distance (AVGDIST). The confusion matrix is shown in Figure 2.

By employing these metrics, we gain a comprehensive evaluation of the model’s performance, providing a detailed analysis and comparison of its predictive accuracy and segmentation quality. This comprehensive evaluation ensures a thorough assessment of the model’s performance.

When calculating accuracy, TN, which represents correctly identified background regions, is included in the calculation. Since the background makes up the majority of the Ground Truth, the Accuracy metric tends to yield better results compared to other metrics. The formula for calculating accuracy is
(1)$$\mathrm{Accuracy}=\frac{TP+TN}{TN+FP+TP+FN}$$

The Intersection over Union (IoU) metric overlays the predicted results and the Ground Truth and then calculates the intersection and union of the obtained images. For medical images with substantial black regions, a modified IoU formula for evaluating the training area may be used. The specific IoU formula is expressed by
(2)$$\mathrm{IoU}=\frac{TP}{FP+TP+FN}$$

The Dice Score Coefficient (DSC) is commonly used to calculate segmentation images. This metric measures the similarity between two samples, with values ranging from 0 to 1, where 1 indicates the best possible similarity and 0 is the worst. The formula for calculating DSC should be provided for reference, as follows:(3)$$\mathrm{DSC}=\frac{2TP}{FP+2TP+FN}$$

The Average Hausdorff Distance (AVGDIST) is primarily used for edge detection in segmentation images. This metric is used to evaluate the distance between the segmentation boundaries of predicted and actual results. The formula for calculating AVGDIST is provided as follows:(4)$$\mathrm{Average}\;\mathrm{Hausdorff}\;\mathrm{distance}\;(\frac{GtoS}{G}+\frac{StoG}{S})/2$$
3.Actual Predicted Lesion Segmentation Output:

In addition to quantifying metrics and loss, the actual predicted results are a focal point of this study. We overlay the predicted lesion segmentation results on the original CT images and compare them with manually annotated lesion areas. This comparison highlights the differences between the model’s predictions and the actual lesion annotations. The predicted lesion segmentation results are illustrated in Figure 3.

By examining the predicted lesion segmentations against the actual annotations, we can evaluate the model’s performance in accurately identifying and segmenting pulmonary lesions. This analysis provides insights into the model’s strengths and weaknesses in predicting various lesion types. It enables a comprehensive understanding of the model’s efficacy and reliability in clinical settings.

### 2.4. Model Framework

U-Net, proposed by Olaf Ronneberger et al. in 2015 [12], was specifically designed to address challenges in medical image segmentation. A key feature of the U-Net architecture is the use of skip connections, which are intended to effectively capture and utilize features for segmentation. U-Net can be divided into two parts: the encoder and the decoder. The left half of the network is the encoder, which is used for feature extraction, while the right half is the decoder. The upsampling method is used to gradually increase the extracted features to roughly match the input size, thus mitigating feature loss that occurs during encoding and decoding. U-Net performs well with small datasets, making it a suitable choice for situations where medical imaging data is difficult to obtain. Therefore, we have chosen to use U-Net as the foundation for our approach.

In 2018, Oktay et al. [13] proposed the Attention U-Net model for medical image segmentation. The main difference between Attention U-Net and the conventional U-Net lies in adding the Attention Gates (AGs) module (as shown in Figure 4). The AGs module receives the feature maps from the original input image. The input image is first processed separately using convolutional layers to consolidate its spatial dimensions. Then, the feature maps and the original input image are combined using element-wise addition (Add operation) to emphasize the regions of interest. The AGs module is designed to enhance and highlight the relevant and salient features in the image while suppressing irrelevant areas. This selective attention mechanism allows the model to focus on important regions and effectively suppress noise or non-relevant information, leading to improved accuracy and reduced overfitting. The use of the ReLU function further aids in preventing overfitting during training. By training the model with AGs, it learns to prioritize and concentrate on the informative features, leading to better segmentation results. Additionally, the attention mechanism helps to control computational complexity, making the training process more efficient. The overall architecture of the Attention U-Net is depicted in Figure 5.

The Dense Block (DB) module possesses the ability of feature reuse by employing Concatenation (Concat) in its forward propagation process, which connects all layers in a manner that reduces parameter usage through dimension Concatenation. This fusion of features helps mitigate the vanishing gradient problem. The structure of the DB module is shown in Figure 6.

The Residual Block (RB) uses shortcuts to connect input and output through Add in the graph. After equivalent mapping across intermediate layers, it does not generate additional parameters or increase computational complexity. This alleviates the neural network by skipping but preserving features and thus reduces the overall computational time. The structure is shown in Figure 7.

The RDAG U-Net model structure is based on the U-Net architecture. It combines RB and DB in the encoder part to reduce training time while maintaining effective feature extraction. Additionally, the model employs AGs to suppress non-relevant regions and further enhance its performance. The RDAG U-Net model structure is shown in Figure 8. RDAG U-Net consists of two main parts: the encoder and the decoder. In the encoder, the image undergoes five image-processing steps, including two RBs and three DBs. Each RB is followed by a pooling layer, and each DB is followed by a convolutional layer and a pooling layer. The convolutional layers are responsible for capturing image features, while the pooling layers down-sample the image and reduce its size.

To preserve image features to the fullest extent, the output of each layer in the encoder is connected to the corresponding decoder layer through skip connections (blue dashed arrows). This process ensures that the feature maps from the encoder are combined with the feature maps from the same decoder level. Additionally, AG modules are placed in the decoder to focus on critical salient features and suppress unnecessary regions. In the right half of the decoder, the feature maps obtained from AGs are concatenated and passed through a 3 × 3 convolutional module (Conv. + ReLU). It is further split into two parts. The first part employs a second 1 × 1 convolutional module (Conv. + Batch Normalization + ReLU) to adjust the image resolution to the same level as the skip connections. It is then passed through AGs to suppress irrelevant regions and concatenated with the 3 × 3 convolutional module. The second part involves up-sampling using the 3 × 3 convolutional module, and the up-sampled image is concatenated with AGs and further processed by a 3 × 3 convolutional module for restoration.

The decoder aims to reconstruct the feature maps obtained from the encoder to achieve the exact resolution and pixel values as the original input image. This process enables the decoder to restore the feature maps to their original dimensions, obtaining results with the exact resolution and pixel values as the input image.

### 2.5. Training Environment Setup 

The model was constructed and trained on a computer. Due to the use of medical images, which have high resolution, there are requirements for GPU memory capacity. If the original images were used with a resolution of 512 × 512 pixels, they would not be operable due to hardware limitations. Therefore, the images were resized to 224 × 224 pixels for training. The programming was conducted in Python, and we ensured compatibility between the GPU we used and the software versions. The training was performed on a computer with the specifications listed in Table 2.

## 3. Results and Discussion

### 3.1. Interpolation Method Selection

In the interpolation [21] experiments, the following five interpolation methods were used:(1)Area interpolation [30]: Also known as “Pixel Area Resampling”, this method calculates the average pixel value within each target pixel’s area. It is useful when resizing images to reduce the blocky artifacts that may occur with other methods like nearest neighbor interpolation;(2)Linear interpolation [31]: Linear interpolation uses a straight line between two adjacent data points to estimate the value at the target point. It is computationally efficient and suitable for smoothly varying data;(3)Nearest interpolation [32]: In nearest interpolation, the value of the nearest data point to the target point is assigned as the estimated value. It is a simple and fast interpolation method but may lead to blocky artifacts and might not capture smooth changes in data;(4)Lanczos interpolation [33]: Lanczos interpolation is a high-quality interpolation method that uses a sinc function to estimate values at non-grid points. It provides sharp details and reduces aliasing artifacts.

The four interpolation methods were tested preliminarily, and area and nearest interpolation methods were selected based on the validation accuracy results for further testing. The test results are recorded in Table 3 as follows.

After observing the experimental results, it was found that nearest interpolation exhibits the characteristic of sharp edges without transitional regions. Considering that ground-glass opacities often appear as hazy and tiny clusters, using area interpolation indeed preserves the original appearance of the image. However, from a computer’s perspective, well-defined boundaries represent stronger feature intensities that can be obtained through convolution (as demonstrated in the comparison shown in Figure 9 and Figure 10).

To validate this statement, we employed the IOU loss to calculate the difference between the neural networks trained using area and nearest interpolation. We found that the model trained with area interpolation is more prone to feature loss during the training process due to the blurred edges of organs and lesions. As seen in the data from Table 3, the IOU loss of the model trained with area interpolation is significantly higher than that of the model trained with nearest interpolation. Therefore, we decided to use nearest interpolation as the interpolation method for model training.

We found that the IOU loss of the model trained with area interpolation was significantly higher than that of the model trained with nearest interpolation. As a result, we decided to use nearest interpolation as the interpolation method for model training.

### 3.2. Data Augmentation and HU Value Adjustment Module

To facilitate more efficient deep learning training, a substantial database is required. However, due to the challenges in obtaining medical imaging data, data augmentation is employed to expand the training dataset. In this study, five data augmentation techniques were utilized, including size scaling, angle rotation, pixel shifting, image cropping, and horizontal flipping. By using this module, our training data was expanded from 3520 images to 10,560 images. Figure 11 illustrates the original images enhanced by our data augmentation module. This augmentation process enhances the diversity of the training dataset and improves the model’s performance.

HU value adjustment can enhance the contrast between lesion and non-lesion areas in medical images, improving the model’s performance. It can also enhance the visibility of lesions in medical images taken without contrast agents. The images before and after HU value adjustment are shown in Figure 12.

### 3.3. Training Time and Convolution Parameter Comparison

After conducting the experiments, we also compared the training time and convolution parameters between different models. Compared to the convolutional networks Attention U-Net [13] and Attention Res U-Net [34], our convolutional network, RDAG U-Net, achieves excellent accuracy with lower time costs. Additionally, when compared to Attention Res U-Net [34], our model has more convolution parameters. The RDAG U-Net model effectively utilizes ResBlocks with skip connections and combines them with DenseBlocks to reuse important features, leading to rapid convergence to high accuracy. The training time for RDAG U-Net is 54 s, for Attention U-Net it is 98 s, and for Attention Res U-Net it is 57 s. Overall, the training time of RDAG U-Net is approximately 45% lower than that of Attention U-Net. The convolutional parameters and training times are shown in Table 4 below.

### 3.4. Training Convergence Curve

Using 20 lung CT images obtained from TCGA, 3520 images were processed, standardizing all image pixels to 224 × 224. Subsequently, the images underwent HU value restriction and data augmentation, increasing the dataset size to 10560 images. The Adam optimizer was used, with a learning rate of 10^−4^ for the RDAG U-Net model. The batch size was set to 7, and training was conducted for 120 epochs. The results were very satisfactory, with an ACC value of 0.9329 for the lesions, as shown in Figure 13. In the figure, the training curve and validation curve are very close, and the training curve does not exceed the validation curve, indicating no signs of overfitting for the model.

The loss function indicates how poorly the model is performing, representing the degree of misfit of the current data. Therefore, a lower loss value indicates better model performance. In this experiment, the developed model achieved a loss value of 0.095, as shown in Figure 14, which depicts the training result curve for the loss value. The validation loss curve in the figure does not show an increase, indicating that the model is not experiencing overfitting.

### 3.5. Lung Lesion Recognition Results

In this chapter, we conducted a comparison of the segmentation capabilities among three convolutional models: RDAG U-Net, Attention U-Net, and Attention Res U-Net. As shown in Table 5, the RDAG U-Net model achieved the highest accuracy and the lowest loss values in both the training and validation sets. It is noteworthy that, while its overall performance is only slightly lower in terms of Dice Similarity Coefficient (DSC) compared to Attention U-Net, the remaining results are superior to the other two models, making it the overall best-performing and most stable model. Additionally, when compared to Attention U-Net [13] and Attention Res U-Net [34], the RDAG U-Net model requires the least training time to achieve satisfactory training results.

### 3.6. Analysis of Lung Lesion Results Using RDAG U-Net

This section will showcase the actual results of the model predictions. In addition to presenting the outcomes from the data, this aims to demonstrate the model’s practical predictive capabilities and assess its applicability in real-world scenarios. The model’s prediction results are displayed in Figure 15.

Due to the highly irregular nature of lung lesion patterns, they exhibit a diverse range of shapes and sizes. To assess the model’s actual predictive performance, we selected cases with large lesions, small lesions, and mixed-size lesions for evaluation.

(a) Large lesion: The ACC reached 91.2%, and it can be seen from the prediction image that the main large lesions are well predicted, with edges very close to the Ground Truth. A small lesion on the left has also been successfully predicted, as shown in Figure 15a.

(b) Small lesion: The model achieved an ACC of 92.6% when predicting minor lesions and was able to accurately predict their positions and contours, as shown in Figure 15b.

(c) Mixed large and small lesions: The model achieved an ACC of 90.4% in these cases and effectively predicted irregular lesions and minor lesions in the upper part, as shown in Figure 15c.

(d) Unilateral lesions: In cases of unilateral lesions, the RDAG U-Net model was not affected by the healthy side and effectively predicted the presence of lesions, with an ACC of 91.5%. The prediction image is shown in Figure 15d.

### 3.7. Evaluation and Comparison of Mild and Severe Pneumonia Models

We divided our database into mild and severe cases and compared the results of three different models: RDAG U-Net, Attention U-Net, and Attention Res U-Net. The database classification is shown in Figure 16.

For lung lesion segmentation, we compared the results of the three different convolution models (Attention U-Net, Attention Res U-Net, and our RDAG U-Net) in two different cases, as shown in Figure 17. In the severe case, all three convolution models achieved better training results. However, when dealing with mild cases, due to the presence of more small and dispersed lesions, there is a tendency for a decline in model prediction performance. From the results, the performance of all three convolutional models significantly decreased. Although the performance of Attention Res U-Net was better than our RDAG U-Net in severe cases when the data transitioned to mild cases, the performance of Attention Res U-Net decreased from 93.6% to 63.6%, Attention U-Net decreased from 85.4% to 67.4%, and RDAG U-Net decreased from 90.4% to 85.6%, only dropping by about five percentage points. This indicates that the model still performs well when faced with fewer ground truths. As we divided the original database into two parts, this also demonstrates the model’s learning ability on a small database. In the figure, it can be observed that our model maintains good accuracy in both mild and severe conditions.

In Figure 18, the analysis of the DSC is conducted for different severity cases using Attention U-Net, Attention Res U-Net, and our RDAG U-Net, as illustrated. This computation assesses the similarity between the Ground Truth and the predicted results. From Figure 18, it is observed that for severe lesions, RDAG U-Net slightly outperformed Attention U-Net with 79.8% compared to 79.3%. However, when dealing with more fragmented mild cases, RDAG U-Net decreased to 72.6%, Attention U-Net decreased to 53.1%, and Attention Res U-Net decreased to 55.2%. Despite the slight decrease in DSC for RDAG U-Net, it maintained high stability compared to the other two models. 

### 3.8. Discussion of Experimental Results

The RDAG U-Net shows notable advantages in segmentation performance, achieving the highest accuracy and the lowest loss value among the three models, and also having the shortest training time. As shown in Figure 15, the RDAG U-Net model can accurately predict and delineate lesions that are not visible in the original CT images. Although there are occasional segmentation errors for smaller and more dispersed lesions, this does not impact the prediction of major lesions. The model’s performance demonstrates its capability to detect and segment various types of lung lesions, making it a valuable tool for practical medical applications. Overall, the RDAG U-Net outperforms the other two models in terms of stability and performance.

### 3.9. Lung Predictive 3D Visualization Model

After predicting 2D lesion images through the model, the predicted images are imported into the ITK-Snap software (ITK-Snap 3.6.0) to generate a 3D model of lesion and lung segmentation. This 3D visualization model provides medical professionals with a representation closer to real-life lesion scenarios. It allows the identification of the actual position of the lesions by adjusting angles and sizes and enhances clarity by adjusting the transparency of layers. Additionally, since the original CT image files include the thickness of each slice, the distances and positions of the 3D model can better approximate real-world situations. The process diagram of the 3D transformation and the 3D visualization model are shown in Figure 19.

## 4. Conclusions

This research leveraged three models for the automatic segmentation of pneumonia lesions in CT images: the publicly available Attention U-Net [13], Attention Res U-Net [34,35], and the novel RDAG U-Net. The RDAG U-Net model was designed with Dense Blocks in the encoder to facilitate feature reuse and mitigate gradient vanishing, while Res Blocks with skip connections were incorporated to reduce training time. In the decoder, Attention Gates (AGs) and specialized convolution methods were combined to focus on relevant features and ignore irrelevant regions, effectively controlling computational complexity. A total of 3520 CT images were collected from open-source websites and augmented to increase the dataset to 10,560 images. The key findings of this research include the following:HU Value Modification: Adjusting the HU values allowed CT images to display lesions more clearly, even without the use of contrast agents.Computational Efficiency: RDAG U-Net demonstrated the fastest computational speed among the three models, reducing computation time by approximately 45% compared to Attention U-Net.Accuracy: Utilizing data from open-source images (TCGA), RDAG U-Net achieved an accuracy of 93.29% in pneumonia lesion identification.

The strengths of RDAG U-Net lie in its rapid and accurate performance, making it particularly advantageous for the diagnosis of respiratory diseases such as COVID-19. Its ability to differentiate CT images clearly without the need for contrast agents also benefits patients who cannot tolerate the side effects of such agents. Moreover, the model’s versatility extends to various medical imaging formats, including MRI and ultrasound, enhancing its applicability across diverse clinical settings.

In conclusion, RDAG U-Net offers a significant advancement in medical imaging diagnostics by providing a fast, accurate, and non-invasive method for identifying pneumonia lesions. This technology has the potential to greatly benefit both patients and healthcare providers by improving diagnostic accuracy and reducing the workload on medical professionals. Future work will focus on further refining the model and exploring its applications in other areas of medical imaging. This improved conclusion emphasizes the technical contributions, practical applications, and future potential of the RDAG U-Net model, making it compelling for reviewers and relevant to the field of medical imaging research.

## Figures and Tables

**Figure 1 diagnostics-14-02099-f001:**
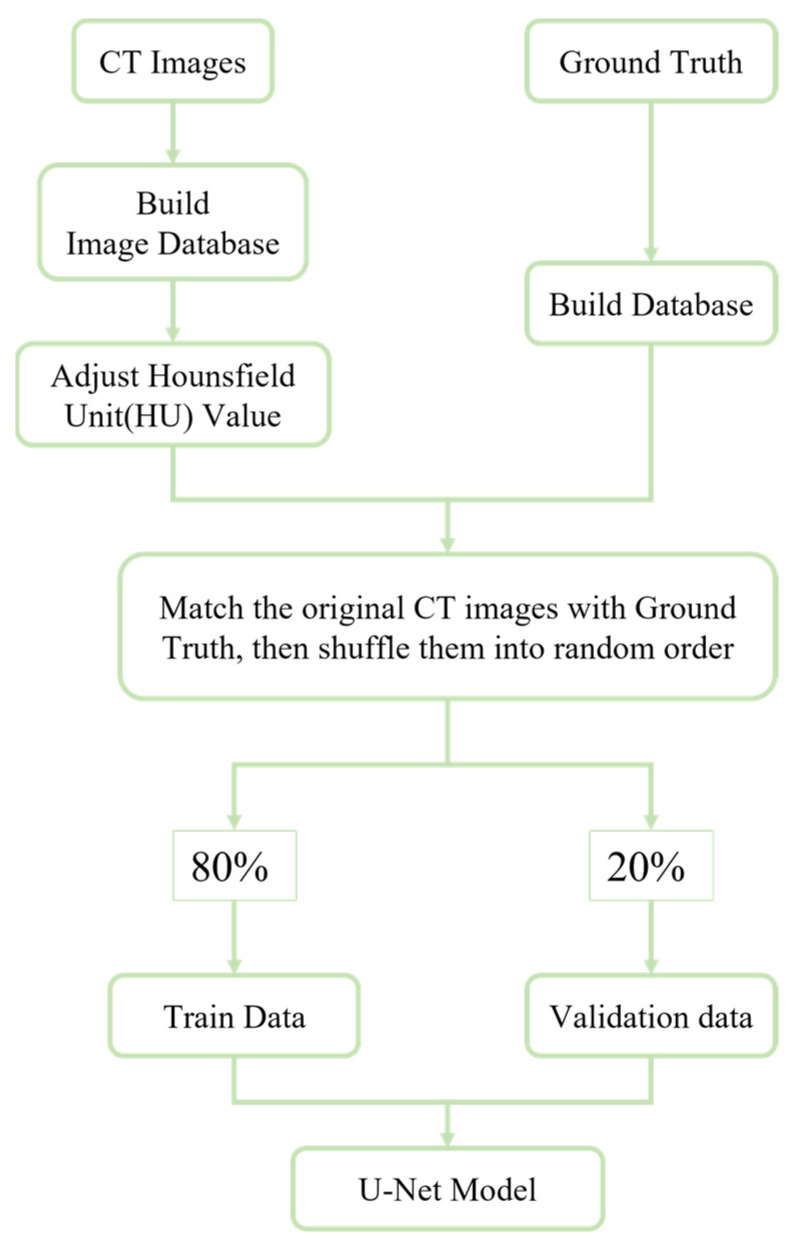
Data and configuration workflow diagram.

**Figure 2 diagnostics-14-02099-f002:**
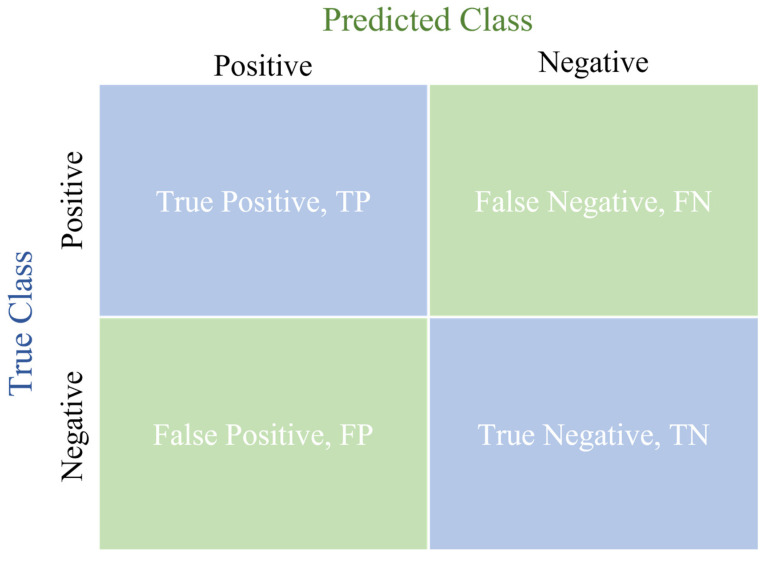
Confusion matrix.

**Figure 3 diagnostics-14-02099-f003:**
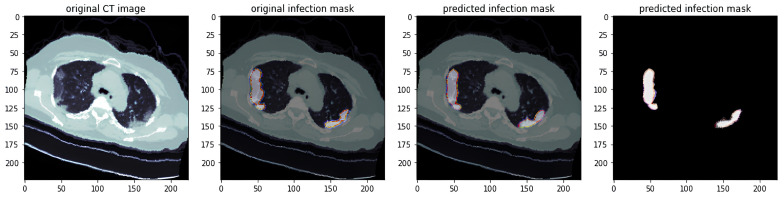
Lesion prediction and segmentation results (The horizontal and vertical coordinates are used to identify an image with dimensions of 224 × 224).

**Figure 4 diagnostics-14-02099-f004:**
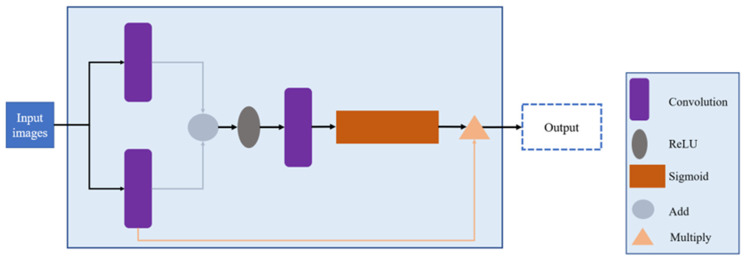
Attention Gates (AGs) module.

**Figure 5 diagnostics-14-02099-f005:**
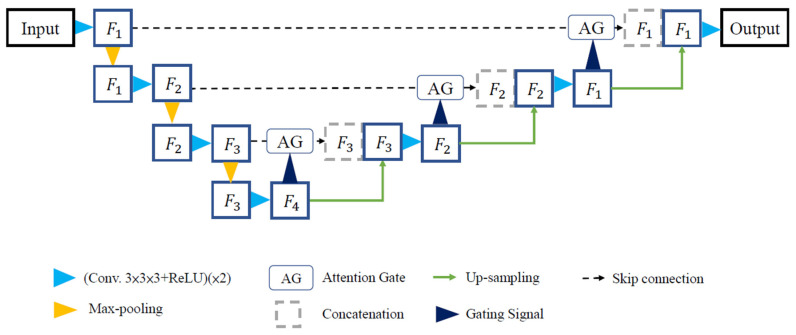
Attention U-Net model architecture.

**Figure 6 diagnostics-14-02099-f006:**
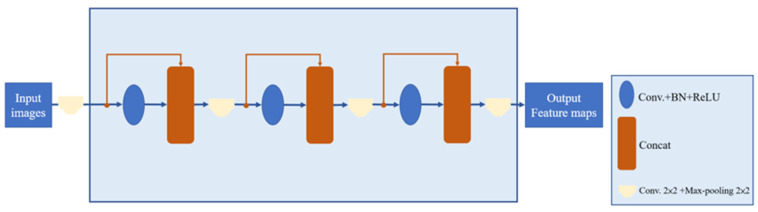
Dense Block module internal structure.

**Figure 7 diagnostics-14-02099-f007:**
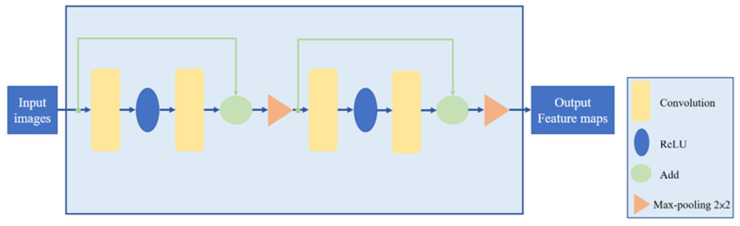
Res Block module internal structure.

**Figure 8 diagnostics-14-02099-f008:**
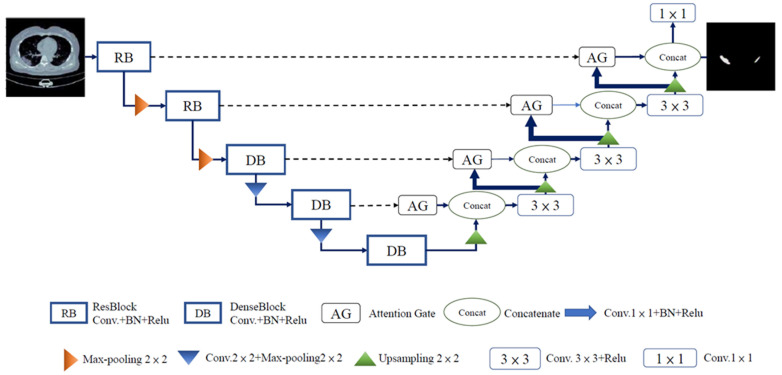
RDAG U-Net model.

**Figure 9 diagnostics-14-02099-f009:**
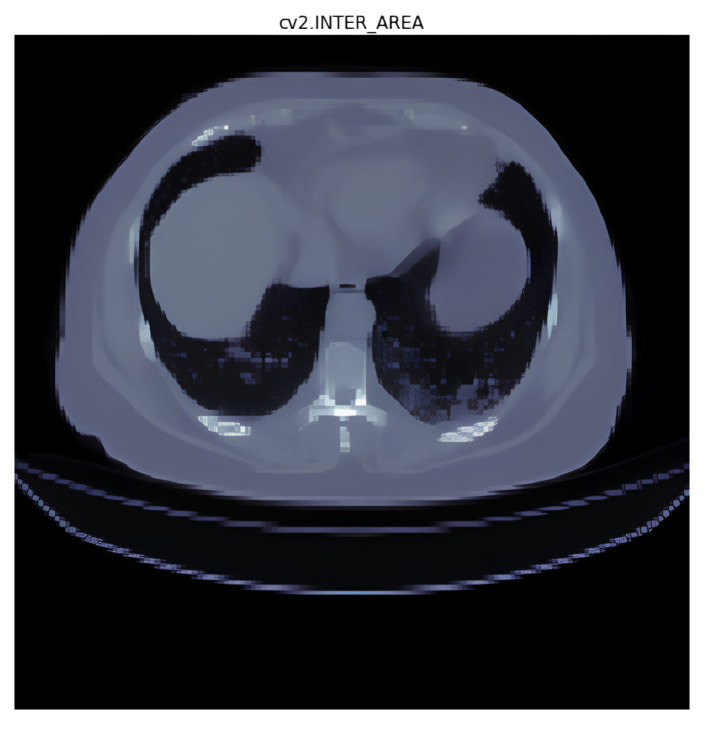
Area interpolation (cv2.INTER_AREA: the name of the interpolation method used).

**Figure 10 diagnostics-14-02099-f010:**
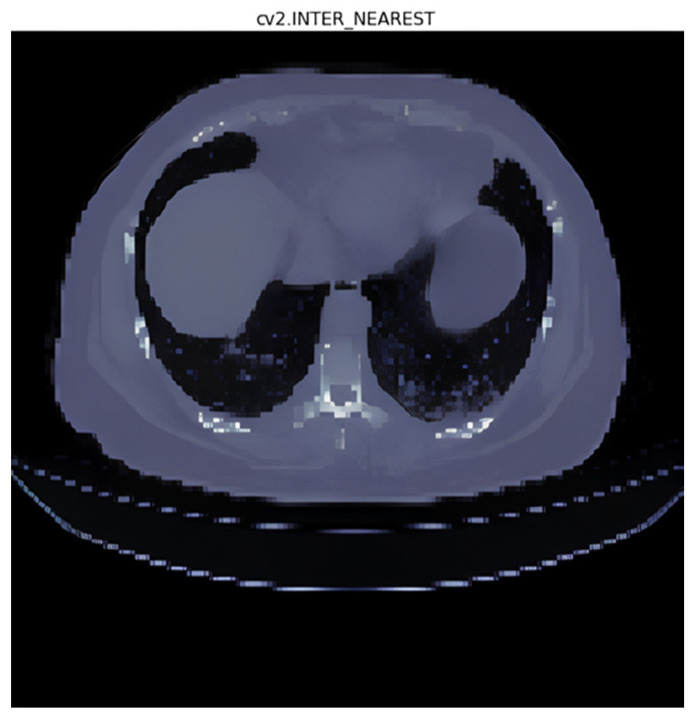
Nearest interpolation (cv2.INTER_NEAREST: the name of the interpolation method used).

**Figure 11 diagnostics-14-02099-f011:**
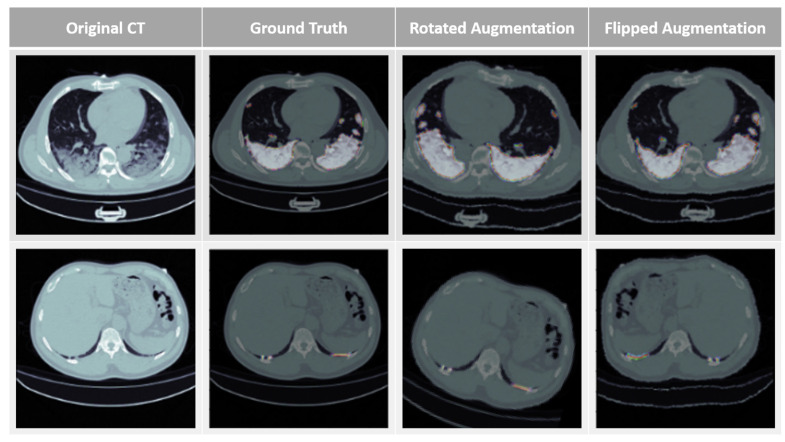
Data Augmentation (from left to right: original CT image, lesion annotation, rotation augmentation, horizontal flip).

**Figure 12 diagnostics-14-02099-f012:**
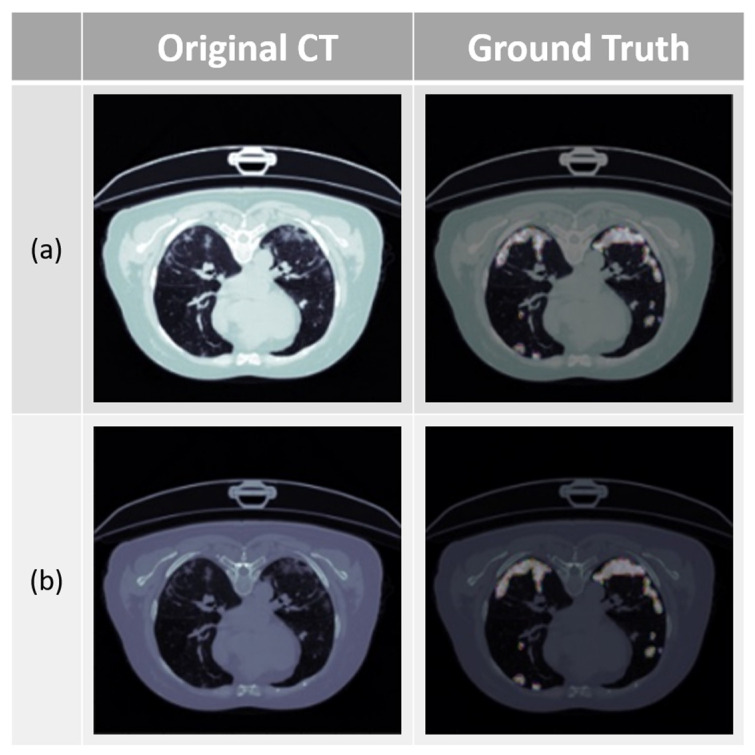
Before and after HU value adjustment ((**a**): before adjustment; (**b**): after adjustment).

**Figure 13 diagnostics-14-02099-f013:**
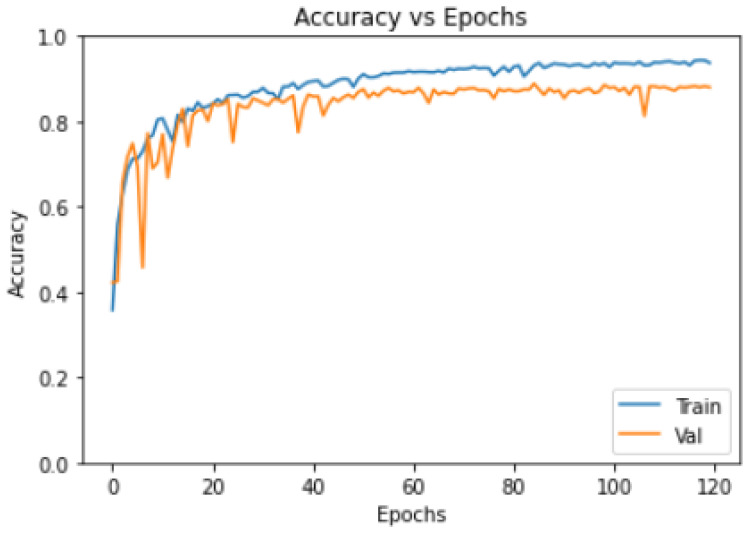
Training accuracy results of 20 patients.

**Figure 14 diagnostics-14-02099-f014:**
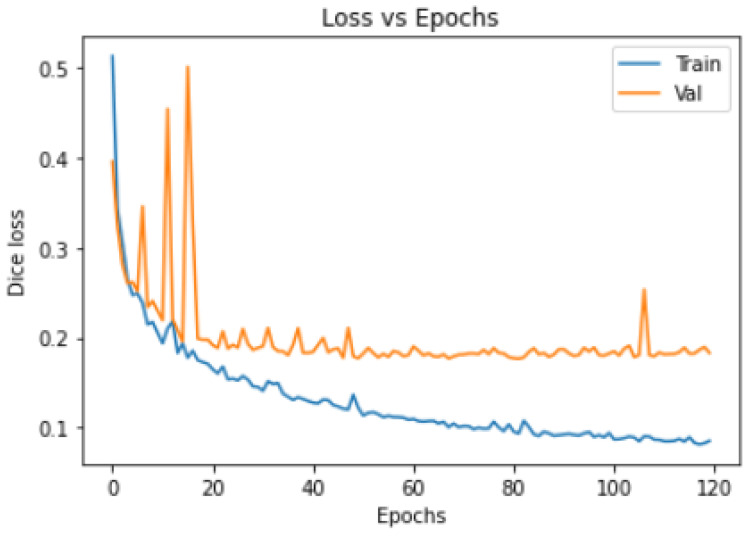
Training loss results of 20 patients.

**Figure 15 diagnostics-14-02099-f015:**
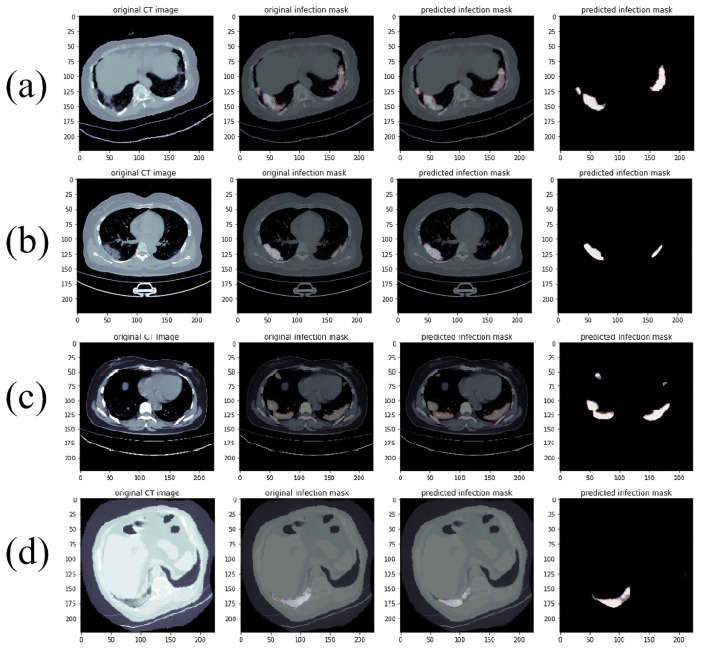
Results of segmentation of lung lesions in RDAG U-Net model. From left to right, original CT images, Ground Truth, the prediction results, and the superposition comparison of Ground Truth and prediction results. (**a**) Large lesion, (**b**) smaller lesion, (**c**) mixed large and small lesions, and (**d**) unilateral lesion (The horizontal and vertical coordinates are used to identify an image with dimensions of 224 × 224).

**Figure 16 diagnostics-14-02099-f016:**
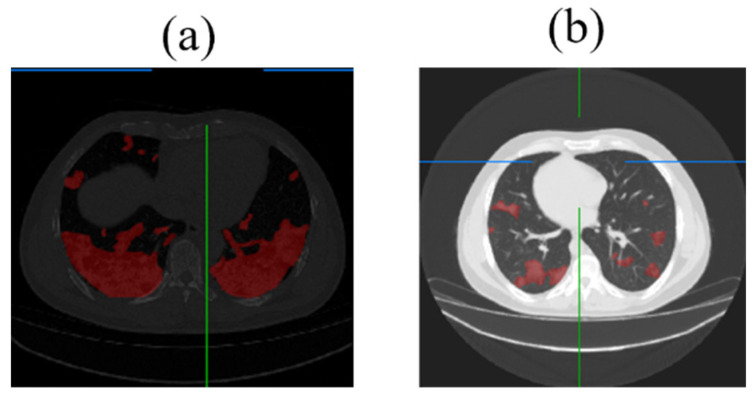
COVID-19 Database Severity Classification (The blue and green lines represent crosshairs for the alignment area, while the red area indicates the lesion). (**a**): severe and (**b**): mild.

**Figure 17 diagnostics-14-02099-f017:**
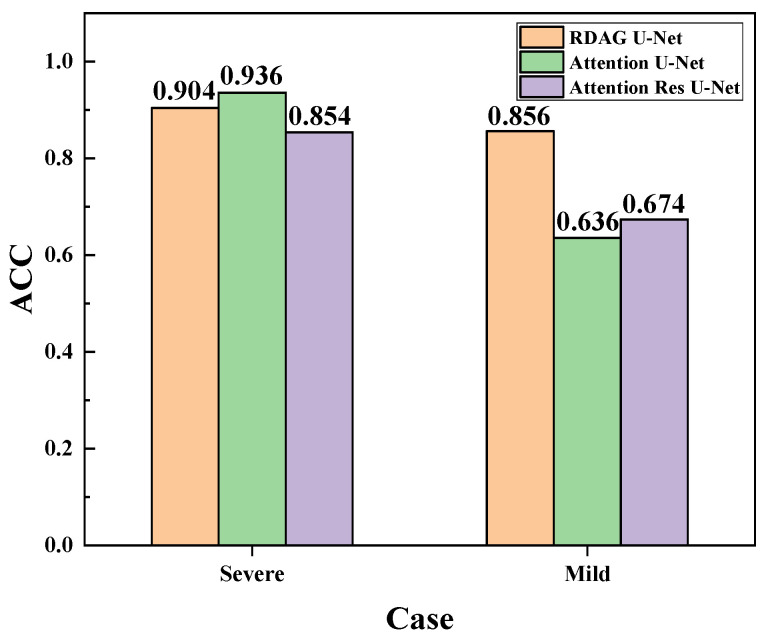
Comparing the ACC of lesion segmentation results for two scenarios using three models.

**Figure 18 diagnostics-14-02099-f018:**
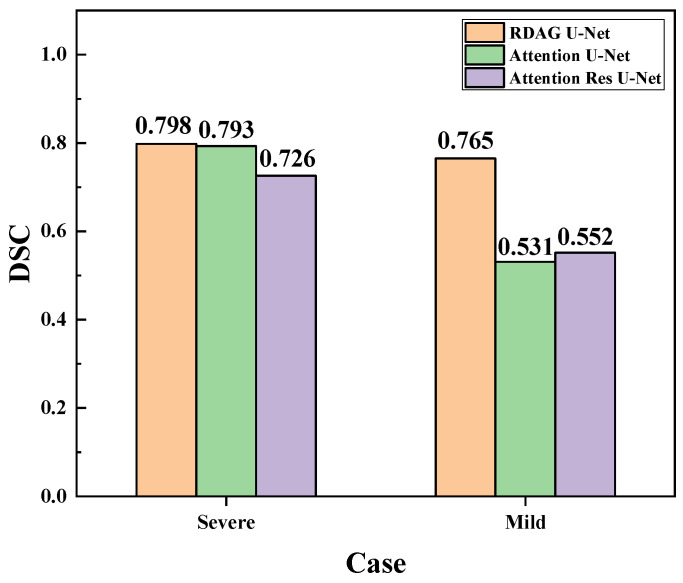
Comparing the DSC of lesion segmentation results for two scenarios using three models.

**Figure 19 diagnostics-14-02099-f019:**
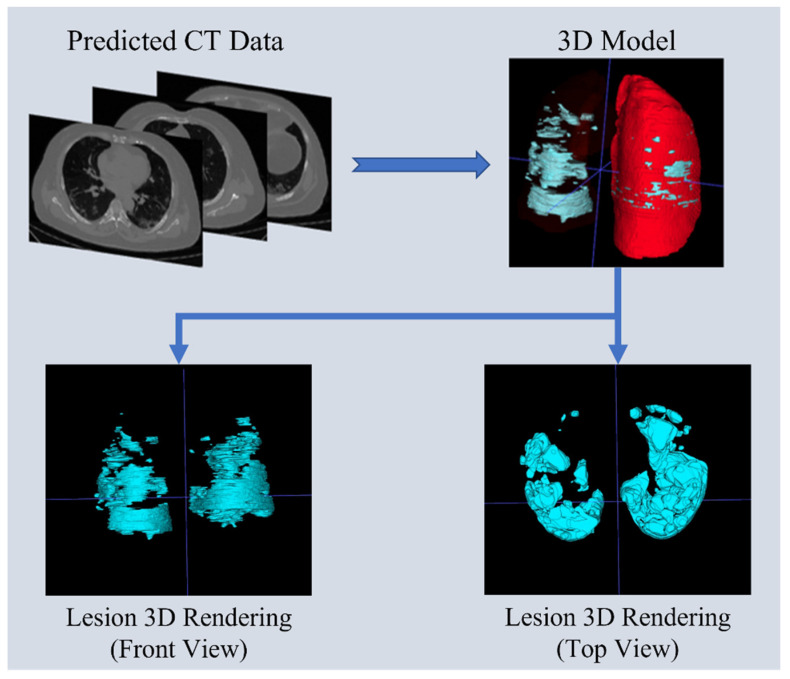
Three-dimensional visualization model transformation process diagram and the segmentation model for lungs and lesions (The blue lines represent the crosshairs for the alignment area. In the 3D model, the red areas represent the entire lung, while the blue areas represent the lesion in the 3D model).

**Table 1 diagnostics-14-02099-t001:** The features of various models.

Reference	Model	Disease	Features
[13]	Attention U-Net	Pancreatic disease	Developed Attention U-N to enhance the performance of the traditional U-NET
[14]	ASPP U-Net	Retinal vessels	Captures contextual information from different scales, improving the segmentation of complex image structures
[15]	U-Net with other classification models	COVID-19	Classifies lesions as COVID-19 based on segmentation results
[16]	TB-Net	Tuberculosis	Specifically designed for detecting tuberculosis that is challenging to identify with traditional visual inspection
[17]	Dual Kmax UX-Net	sub-regions of organs	Uses information of unlabeled samples to determine labels, improving the situation of insufficient labeled data
Our work	RDAG U-Net	SARS-CoV-2 Pneumonia	Builds on U-Net by adding multiple modules to enhance accuracy, reduce training time, and successfully predict lesions without contrast agents by improving HU values

**Table 2 diagnostics-14-02099-t002:** Computer specifications and software versions.

Operating System	Windows 10
CPU	Intel i7 9700KF
GPU	NVIDIA RTX 2080Ti 11GB
SSD	M.2 (PCIe) 512GB*2
RAM	DDR4 128G
Programming Language	Python 3.7.7
Development Environment	Tensorflow-gpu 1.14 Keras 2.3.0

**Table 3 diagnostics-14-02099-t003:** Interpolation methods training result.

	Train Loss	Train Accuracy	Validation Loss	Validation Accuracy
Area	0.088	0.928	0.180	0.891
Linear	0.104	0.913	0.194	0.867
Nearest	0.090	0.930	0.097	0.929
Lanzcos4	0.087	0.943	0.197	0.872

**Table 4 diagnostics-14-02099-t004:** Comparison of model parameters and training time.

Method	Parameter	COVID-19 Lesion-Training Time (s)
RDAG U-Net	13,043,613	54
Attention U-Net [13]	50,952,981	98
Attention Res U-Net [34]	12,975,805	57

**Table 5 diagnostics-14-02099-t005:** Evaluation of lung lesion segmentation results across different models.

Model	ACC	IoU	DSC	AVGDIST
RDAG U-Net	93.29%	73.49%	82.01%	0.227
Attention U-Net [13]	90.10%	69.50%	83.11%	0.214
Attention Res U-Net [34]	91.35%	71.38%	80.15%	0.259

## Data Availability

The raw data supporting the conclusions of this article will be made available by the authors upon request.

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
