# Peer review of "RDAG U-Net: An Advanced AI Model for Efficient and Accurate CT Scan Analysis of SARS-CoV-2 Pneumonia Lesions"

_diagnostics, 2024, doi:10.3390/diagnostics14182099_

Round 1
Reviewer 1 Report
Comments and Suggestions for Authors
need FPGA implementation of the algorithms and compare performance with CPU implementations
Reviewer 2 Report
Comments and Suggestions for Authors
This study demonstrates that the RDAG U-Net model is a significant advance for CT image analysis of respiratory infections such as SARS-CoV-2 pneumonia. The high accuracy and computational efficiency make the model a valuable tool for medical diagnosis and treatment planning. However, the generalizability of the model must be proven. The following comments are important for the development of the article:
1) The performance of the model on different datasets and disease types could be more satisfactory.
2) Section 4 is very general information. I think it should be removed.
3) Previous studies on model development are rarely mentioned in the article. A larger section should be prepared, and the features of previous studies should be shown in a table.
4) Confusion matrices are theoretically explained, but confusion matrices are not used for the results.
5) Increasing the dataset is necessary for model performance. However, in this case, overfitting may occur. Too much data augmentation was done in this study. Did this create an overfitting problem?
6) The discussion section should be prepared separately.
Reviewer 3 Report
Comments and Suggestions for Authors
The statement that there were "650,000 deaths" as of September 2022 seems inaccurate or outdated.
The claim of "up to 45% reduction in training time" with the RDAG U-Net compared to other models is significant but lacks context in the introduction. It's unclear what baseline or conditions are being used for this comparison.
No detailed information on the specific hardware used.
The discussion about the trade-off between resolution and batch size is too general.
The decision to reduce the resolution from 512x512 to 224x224 pixels to enhance training efficiency is not sufficiently justified.
While data augmentation is mentioned, the specific techniques used are not detailed.
A more in-depth explanation of why these specific models were chosen over others and what the expected benefits of each architectural modification are needed.
The discussion of IOU loss being higher for Area interpolation lacks depth.
The discussion notes a decline in performance for mild cases but does not deeply explore why this occurs or how it could be mitigated.
The paper does not provide a quantitative evaluation of the accuracy or utility of the 3D visualization models.
Analysis of misclassification cases are needed.
Round 2
Reviewer 2 Report
Comments and Suggestions for Authors
The article is acceptable.
Reviewer 3 Report
Comments and Suggestions for Authors
The manuscript has been sufficiently improved to warrant publication in Diagnostics.